# A NOVEL METHOD TO DETERMINE THE NUMBER OF LATENT DIMENSIONS WITH SVD

## ABSTRACT

Determining the number of latent dimensions is a ubiquitous problem in machine learning. In this study, we introduce a novel method that relies on SVD to discover the number of latent dimensions. The general principle behind the method is to compare the curve of singular values of the SVD decomposition of a data set with the randomized data set curve. The inferred number of latent dimensions corresponds to the crossing point of the two curves. To evaluate our methodology, we compare it with competing methods such as Kaisers eigenvalue-greater-than-one rule (K1), Parallel Analysis (PA), Velicers MAP test (Minimum Average Partial). We also compare our method with the Silhouette Width (SW) technique which is used in different clustering methods to determine the optimal number of clusters. The result on synthetic data shows that the Parallel Analysis and our method have similar results and more accurate than the other methods, and that our methods is slightly better result than the Parallel Analysis method for the sparse data sets.

## 1 INTRODUCTION

The problem of determining the number of latent dimensions, or latent factors, is ubiquitous in a number of non supervised learning approaches. Matrix factorization techniques are good examples where we need to determine the number of latent dimensions prior to the learning phase. Non linear models such as LDA (Blei et al., 2003) and neural networks also face the issue of stating the number of topics and nodes to include in the model before running an analysis over a data set, a problem that is akin to finding the number of latent factors.

We propose a new method to estimate the number of latent dimensions that relies on the Singular Value Decomposition (SVD) and on a process of comparison of the singular values from the original matrix data with those from from bootstraped samples of the this matrix, whilst the name given to this method, Bootstrap SVD (BSVD). We compare the method to mainstream latent dimensions estimate techniques and over a space of dense vs. sparse matrices for Normal and non-Normal distributions.

This paper is organized as follow. First, we outline some of best known methods and the related works in the next section. Then we explain our algorithm BSVD in section 3. The experiments are presented in section 4 and the results and discussion are reported in section 5. And finally conclusion of the study is given in section 6.

## 2 BACKGROUND

The problem of finding the number of latent factors in a data set dates back to early work by Guttman (1954). It extends to a large array of fields including psychology and social science (Bollen, 2002), bioinformatics (Hira & Gillies, 2015), information retrieval (Venna et al., 2010), and of course statistical learning (Hastie et al., 2009, chapter 14). A systematic review on latent variables and its applications can be found in Li & Chen (2016); Kelava et al. (2017); Skrondal & Rabe-Hesketh (2007).

In the following, we briefly explain some of the best known techniques of factor analysis method to decide the number of factors to retain as mentioned in Ledesma & Valero-Mora (2007) such as Kaiser's eigenvalue-greater-than-one rule (K1), Parallel Analysis, Cattell's Scree test, Velicer's

MAP test ( Minimum Average Partial ). Moreover, we outline SW technique which is used in different clustering methods to determine the optimal number of clusters.

## 2.1 K1 or Kaiser's eigenvalue-greater-than-one rule

The K1-Kaiser method was first introduced by Guttman (1954) and later extended and popularized by Kaiser (1960). The method an early strategy used to detect the number of factors to retain by considering. It relies on the eigenvalues of the correlation matrix of the observed factors an stipulates that the number of eigenvalues greater than one corresponds to the number of latent factors to retain.

Despite its simplicity, some researches consider it unreliable (Ledesma & Valero-Mora, 2007; Mumford et al., 2003; Fabrigar et al., 1999). We will nevertheless consider it in our comparison experiments, given that it is a classic method and the first that introduced the use of eigenvectors of the correlation matrix for determining the number of latent factors. Note that a variant of this method was introduced by Warne & Larsen (2014) that showed improvements but still lacked behind PA (§2.2 and MAP (§2.4) and will therefore not be included in the experiment.

## 2.2 Parallel Analysis (PA)

Parallel Analysis is also based on the correlation matrix between the observed factors. It uses bootstrapping on the correlation matrix and then averages the eigenvalues over the bootstrap runs. Eigenvalues greater than the average data set eigenvalue are kept. Mumford et al. (2003); Ledesma & Valero-Mora (2007). This strategy was proposed by John Horn Horn (1965). Warne & Larsen (2014) showed that PA attempt improves over the Eigenvalue-greater-than-one rule. Several researchers found this method appropriate and more accurate in determining the number of factors to retain Montanelli & Humphreys (1976); Mumford et al. (2003). We will see that it PA a close relationship with BSVD and this is corroborated by the closeness of the results.

## 2.3 Cattell's Scree test

Cattell's Scree test, also known as the "elbow" approach, is a graphical representation method to display the number of components to retain. Scree test is a subjective method which sorts the eigenvalues in decreasing order and shows it on the vertical axis; and the number of components in horizontal axis Ledesma & Valero-Mora (2007). In this strategy, we need to find where the Scree happened and the components on the left side of the slope should be retained. Hambleton & Rovinelli (1986) used the Scree test plot and mention that the method fails when elbow cannot be found. Moreover, Stellefson & Hanik (2008) mentions the Scree test as one of the graphical tests to finding the number of latent variables, they specify that researchers should utilize of the ruler to plot a line across the elbow and then they could keep all the components above it. Despite, the various criticism to use this method by Zwick & Velicer (1986) and Hoyle & Duvall (2004), it is one of the most popular methods to find the number of important factors to retain Mumford et al. (2003); Ledesma & Valero-Mora (2007); Raîche et al. (2013).

## 2.4 Minimum Average Partial method (MAP)

MAP approach is based on PCA and relies on the series of partial correlation matrices to define the number of significant factors to retain (Mumford et al., 2003; Ledesma & Valero-Mora, 2007; Zwick & Velicer, 1986; Oconnor, 2000). This approach is introduced by Velicer (1976). In general, statisticians agree that the MAP and PA are the two techniques which are reliable solution to extract the number of factors to retain with the reasonable result Ledesma & Valero-Mora (2007); Mumford et al. (2003); Oconnor (2000).

## 2.5 Silhouette width (SW)

Clustering is generally tought of as a means to reduce the number of data points, but it can be considered for dimensionality reduction technique, namely by using the cluster's centroid and each point's distance from them as a means to define a new space, which was shown to be more effective than SVD under certain conditions (**?**). Therefore, a method to determine the optimal number of clusters can provide another means to determine the dimensionality of a data set.

PAM method is one of the popular technique to automatically determine the optimal number of clusters with the Silouhette (SW) technique Kodinariya & Makwana (2013). The number of clusters computed by SW is associated with the number of latent dimensions in the dataset.

## 3    BOOSTRAPPED SVD (BSVD)

SVD is a well known matrix factorization technique that decomposes the original matrix, $\mathbf{R}$, into the product of two eigenvector matrices, the eigenvectors of the cross-product of the rows and columns, and a of the diagonal matrix of their common singular values.

$$\mathbf{R} = \mathbf{U}_{m \times m} \, \mathbf{\Sigma}_{m \times n} \, \mathbf{V}_{n \times n}^{T}$$

where U and V are orthogonal, and $\mathbf{\Sigma}$ is a diagonal matrix with positive real values.

The singular values represent the importance of the eigenvectors, ordered by decreasing values.

The BSVD method determines the number of dimensions as the point where the singular values of $\Sigma$ cross the singular values $\mathbf{\Sigma}_B$ of a randomized (through bootstrap sampling) matrix $\mathbf{R}_B$. An interpretation of this crossing point where $\mathbf{\Sigma}$ and $\mathbf{\Sigma}_B$ meet is that the remaining singular values are no more due to influential factors, at least in a linear framework.

An example of this can be seen in figure 2 for a data set that was generated using 9 dimensions (vertical line) with uniform distributed values. In this case, it is also easy to tell the number from the elbow at dimensions= 9.

The bootstrapped samples $\mathbf{R}_B$ are simply generated through random sampling with replacement of the values of $\mathbf{R}$.

In the next section, we look at the details of generating data sets and the experiments.

## 4    EXPERIMENTS

We evaluate the ability of BSVD to identify the number of latent dimensions by using synthetic data. Although the use of synthetic data limits the generalizability of our conclusions to the real world by making strong assumptions on the data, it remains the best validation methodology given that we know the ground truth behind the synthetic data and we can control the sparsity and the underlying distributions of each of the observed variables in order to explore this space of conditions.

### 4.1    DATA SETS

#### 4.1.1    SYNTHETIC DATA

The synthetic data is generated by sampling from distributions to create two matrices, $\mathbf{P}$ and $\mathbf{Q}$. Then, $\mathbf{R}$ is obtained by the product $\mathbf{P} \cdot \mathbf{Q}$ plus Gaussian noise.

We use two types of distributions, the normal (Gaussian) with mean= 0 and standard deviation= 1, and the uniform distribution with mean= 2.5 to generate the columns of $\mathbf{P}$ and rows of $\mathbf{Q}$. The choice of 2.5 is inspired from rating-type of data found in recommender systems.

The Gaussian noise added to $\mathbf{P} \cdot \mathbf{Q}$ corresponds to one standard deviation of $\mathbf{R}$ with mean= 0.

All $\mathbf{R}$ matrices are of size $150 \times 240$ and we explore the latent dimensions from 2 to 24.

In figure 1, we illustrate an example of the generated non-normal data set with size of $5 \times 6$ and latent dimension 3.

Data sets of different density are generated, since sparsity is a constraint that we often have to deal with in fields such as recommender systems (rating matrices) and natural language processing (term-document matrices). Sparsity is created by randomly selecting the missing value cells.

In order to capture the behavior of our method when we face a sparse matrix, we employ the algorithm with different percentages of sparseness to the data set (see algorithm 1). Then, we compare the result with the existing mentioned approaches in Table 2. To do so, we follow the next steps for each iteration of latent dimension ($j$):

1) We apply a different percentage of sparseness ($j$) from 10: 90 to our data set with random selection.

2) We impute each missing value by the average of the mean of corresponding row and column.

3) Apply BSVD. And for each latent dimension ($j$); record the result in each iteration of $k$.

4) Compute the average accuracy of each method, when ($j$) terminate.

Figure 5 displays the accuracy of all the methods in the non-normal sparse data set with latent dimension ($j$) equal to 2.

## 4.2 SECOND EXPERIMENT ON THE NORMAL DATA SET

We repeat the previous experiments on the generated simulated random data set with a normal distribution.

## 4.3 ALGORITHMS

The BSVD algorithm is compared with Horn's PA and K1 implementations from Hayton et al. (2004). Moreover, we used of Very Simple Structure(VSS) and PAM packages of R to have the outcome of MAP and SW methods respectively.

## 5 DISCUSSION AND RESULTS

According to the results of provided experiments in the tables 1 and 2, we could show that our method has a better performance than those mentioned especially in the sparse data sets. Our empirical experiments demonstrate that on the dense data sets; the accuracy of BSVD and PA is equal and better than the other approaches. But when we apply a different percentage of sparseness to our data sets, our method is more precise.

In the figures 3 and 4, we display the behavior of each method in the dense and sparse data sets.

Figure 3 depicts the average accuracy of all methods in the dense data sets with normal and non-normal distribution. It shows that MAP method in the dense data set with normal or non-normal distribution has the same accuracy. Additionally, SW technique performs better result with the face of the dense data set with non-normal distribution, while K1 has an extreme behavior in the non-normal data set. Moreover, BSVD, PA and K1 are more precise in the dense data set with normal distribution.

Figure 4 shows the sparse data sets with normal and non-normal distribution. It demonstrates that BSVD, PA, and K1 have better accuracy in the sparse data set with normal distribution but MAP and SW are on the contrary.

Figure 5 shows the average accuracy of all the methods in in different level of sparsity over the non normal sparse data set with latent dimensions (j) equal to 2. The error bars shows the variance of the observations after repeating the algorithm 25 times. Based on the results of these experiments we can conclude that our approach (**BSVD**) is better than the presented methods especially in the sparse data sets. To show if the outcome is statistically significant and is not by chance, we apply t-test between our method and PA. We considered the p values less than or equal to 0.05 as a significant result. To do so, we consider a sample of latent dimensions ($j = \{2, 3, 5, 8, 15\}$) and we repeat twenty-five times the mentioned experiments on the sparse data sets with normal and non-normal distribution, and record the result. Then we apply t-test between BSVD and PA. In this evaluation the null hypothesis (H0) state that

$$\mu_{SVD} = \mu_{PA}$$

and if the H0 is rejected, we could conclude that the obtained results are not by chance and our method is better than PA.

Tables 3 and 4 contain p values of the sparse and dense data sets with normal and non-normal distribution respectively. The first row of each table with 0% of sparsity indicate to the dense data sets.

Table 3 shows more constant behavior, and implies that by increasing sparsity in the sparse data set with normal distribution, BSVD yeilds a significantly better result. But table 4 that shows the result of non-normal sparse data set is hard to interpret. Because the green cells are not affected by increasing sparsity. We can sum up with that the result seems to be significant with increasing the sparsity. In general, according to the tables 3 and 4, the difference between our method and PA seems to be statistically significant by increasing the percentage of sparsity.

## 6 CONCLUSION

The objective of our study was to introduce a new method to find the number of latent dimensions using SVD which we inspired from PA. We employ our method on simulated data sets with normal and non-normal distribution whereas are dense or sparse and compared with the present methods such as PA, MAP, K1, and SW. According to the mentioned experiments and the reported results in the table 1, BSVD and PA have the same accuracy and better than the other presented methods in the dense data sets. But our method has a better result in the sparse data sets which is shown in the table 2. We applied t-test on the sample of latent dimensions (j) between BSVD and PA to demonstrate if the result is statistically significant or not. The results in the tables (3 and 4) demonstrate that in the sparse data sets with increasing the sparsity, our method seems to be significantly better than the other methods.
Our method performance is limited to the presented experiments and data sets. If we want to generalize the method, We need to see the behavior of the algorithm when we have a more complex data set.

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

step a. Generating the matrices x and y with the sizes of $6 \times 3$ and $5 \times 3$.

$$x = \begin{bmatrix} 1 & 0 & 0 \\ 0 & 0 & 1 \\ 0 & 1 & 0 \\ 0 & 0 & 1 \\ 0 & 0 & 1 \\ 1 & 0 & 0 \end{bmatrix}$$

$$y = \begin{bmatrix} 0 & 1 & 0 \\ 0 & 0 & 1 \\ 0 & 1 & 0 \\ 0 & 1 & 0 \\ 0 & 0 & 1 \end{bmatrix}$$

step b. multiply the two matrices to generate M: $M = y \times x^T$.

$$\begin{bmatrix} 0 & 0 & 1 & 0 & 0 & 0 \\ 0 & 1 & 0 & 1 & 1 & 0 \\ 0 & 0 & 1 & 0 & 0 & 0 \\ 0 & 0 & 1 & 0 & 0 & 0 \\ 0 & 1 & 0 & 1 & 1 & 0 \end{bmatrix}$$

step c. we apply $M = 2 \times M + 2$.

$$\begin{bmatrix} 2 & 2 & 4 & 2 & 2 & 2 \\ 2 & 4 & 2 & 4 & 4 & 2 \\ 2 & 2 & 4 & 2 & 2 & 2 \\ 2 & 2 & 4 & 2 & 2 & 2 \\ 2 & 4 & 2 & 4 & 4 & 2 \end{bmatrix}$$

step d. Generating noise to M which is generated with rnorm with mean = 0 and sd = 1 with the same size of M.

$$\begin{bmatrix} 0.46 & 0.40 & 0.70 & -0.62 & 0.43 & 0.55 \\ -1.27 & 0.11 & -0.47 & -1.69 & -0.29 & -0.06 \\ -0.69 & -0.55 & -1.07 & 0.84 & 0.89 & -0.30 \\ -0.45 & 1.78 & -0.22 & 0.15 & 0.87 & -0.38 \\ 1.22 & 0.49 & -1.03 & -1.14 & 0.82 & -0.69 \end{bmatrix}$$

step e. Adding noise to M and bound the values from 1 to 5.

$$\begin{bmatrix} 2 & 2 & 5 & 1 & 2 & 3 \\ 1 & 4 & 2 & 2 & 4 & 2 \\ 1 & 1 & 3 & 3 & 3 & 2 \\ 2 & 4 & 4 & 2 & 3 & 2 \\ 3 & 4 & 1 & 3 & 5 & 1 \end{bmatrix}$$

Figure 1: A Non-normal data set with j = 3

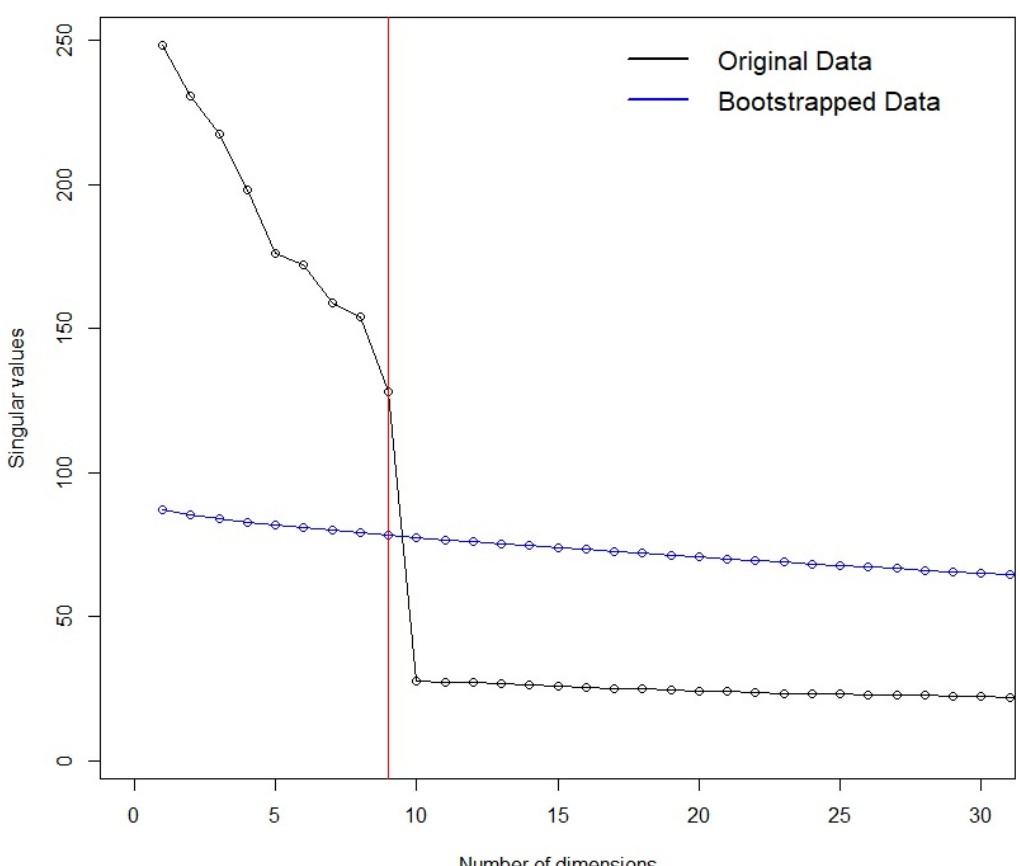

Figure 2: BSVD on a dense synthetic data set with normal distribution.

---

**Algorithm 1** Convert dense data set to sparse for each j

---

    **Input** data set (ds) with latent dimension j
    **Output** the average accuracy of the defined latent dimensions by the algorithm BSVD
1: **procedure** BSVD ON SPARSE DATA SET
2: *loop:* $k \leftarrow 1 : 9$
3:     *d ← (k × 10) × length(ds)/100*
4:     *ind ← sample (length (ds), d)*
5:     *ds [ind] ← NA*
6:     *ds [is.na (ds)]* ← (rowmean + rowcolumn)/2
7:     *result ← BSVD (ds)*
8: **return** *(average accuracy of the result)*
9: **end procedure**

---

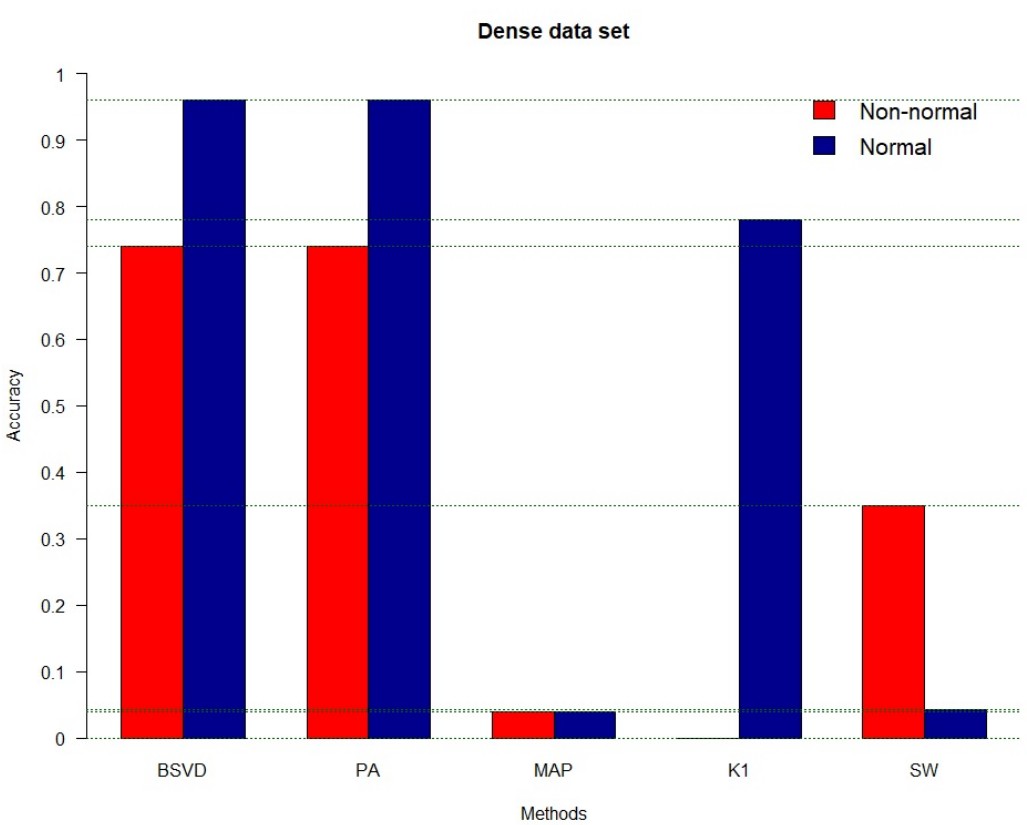

Figure 3: Dense data sets

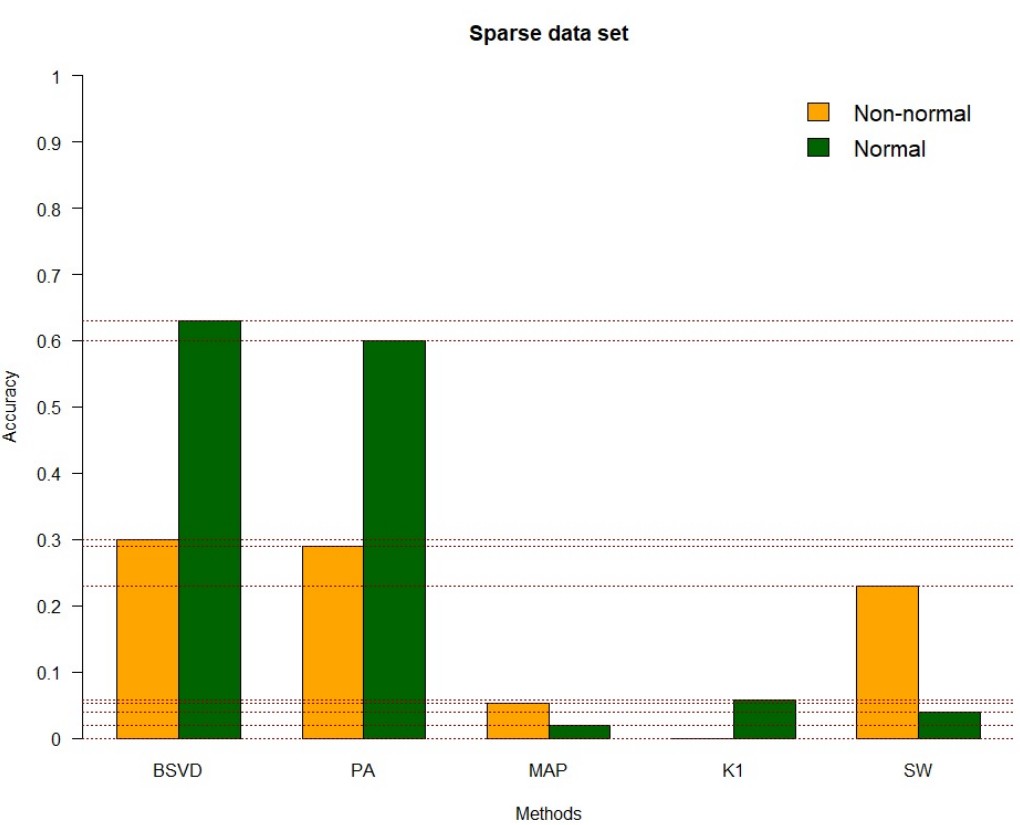

Figure 4: Sparse data sets

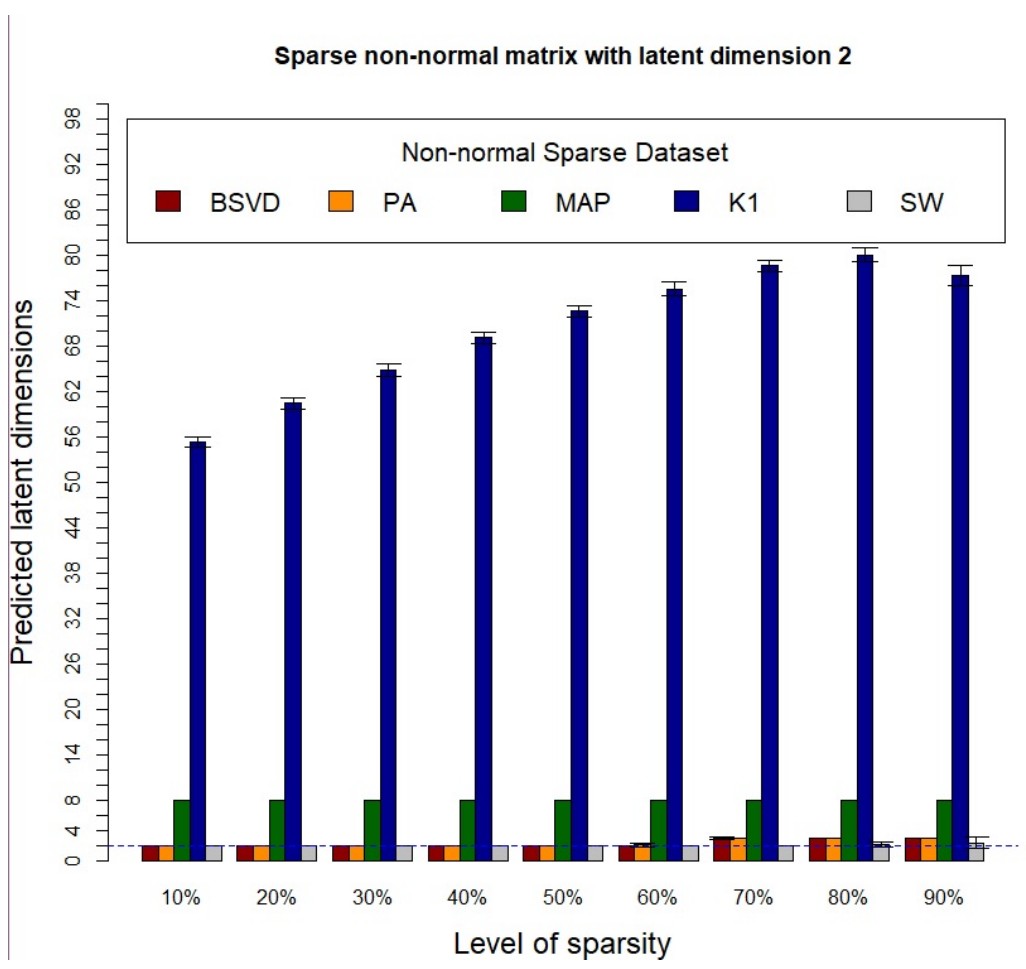

Figure 5: Non-normal sparse data set with j = 2.

Table 1: Result of all the methods in the dense data sets

| Data sets/Methods | BSVD | PA | MAP | K1 | SW |
|---|---|---|---|---|---|
| Non-normal data set | 0.74 | 0.74 | 0.04 | 0 | 0.35 |
| Normal data set | 0.96 | 0.96 | 0.04 | 0.78 | 0.04 |

Table 2: Result of all the methods in the sparse data sets

| Data sets/Methods | BSVD | PA | MAP | K1 | SW |
|---|---|---|---|---|---|
| Non-normal data set | 0.30 | 0.29 | 0.053 | 0.0 | 0.23 |
| Normal data set | 0.63 | 0.60 | 0.19 | 0.58 | 0.39 |

Table 3: Result of p values in the sample of latent dimensions (j) of the sparse data sets with normal distribution

| Sparsity | j = 2 | j = 3 | j = 5 | j = 8 | j = 15 |
|---|---|---|---|---|---|
| 0% | 1 | 1 | 1 | 1 | 1 |
| 10% | 1 | 1 | 1 | 1 | 1 |
| 20% | 1 | 1 | 1 | 1 | 1 |
| 30% | 1 | 1 | 1 | 1 | 1 |
| 40% | 1 | 1 | 1 | 1 | 1 |
| 50% | 0.082 | 0.327 | 1 | 1 | 1 |
| 60% | 0.004 | 0.002 | 0.011 | 0.327 | 1 |
| 70% | 8E-07 | 0.001 | 5E-05 | 0.002 | 0.096 |
| 80% | 8E-08 | 1E-08 | 2E-05 | 1E-05 | 1E-06 |
| 90% | 3E-07 | 8E-08 | 2E-05 | 2E-010 | 1E-08 |

Table 4: Result of p values in the sample of latent dimensions (j) of the sparse data sets with non-normal distribution

| Sparsity | j = 2 | j = 3 | j = 5 | j = 8 | j = 15 |
|---|---|---|---|---|---|
| 0% | 1 | 1 | 1 | 1 | - |
| 10 % | 1 | 1 | 1 | 1 | 0.161 |
| 20% | 1 | 1 | 1 | 1 | 0.185 |
| 30% | 1 | 1 | 1 | 1 | 0.119 |
| 40% | 1 | 1 | 1 | 1 | 0.029 |
| 50% | 1 | 1 | 1 | 0.1034 | 0.032 |
| 60% | 0.327 | 1 | 0.161 | 5E-05 | 3E-04 |
| 70% | 0.327 | 1 | 0.161 | 3E-09 | 3E-06 |
| 80% | 1 | 1 | 0.009 | 1E-06 | 1E-05 |
| 90% | 1 | 1 | 1E-05 | 0.327 | - |

