# OpenReview forum: "A novel method to determine the number of latent dimensions with SVD"
_ICLR.cc/2018/Conference — Reject_

### Official Review · AnonReviewer1 · 2017-11-23
**Poorly described minor variant of rank determination in SVD**

**Rating:** 1
**Confidence:** 4

**Review:**

The manuscript proposes to estimate the number of components in SVD by comparing the eigenvalues to those obtained on bootstrapped version of the input.

The paper has numerous flaws and is clearly below acceptance threshold for any scientific forum. Some of the more obvious issues, each alone sufficient for rejection, include:

1. Discrepancy between motivation and actual work. The method is specifically about determining the rank of a matrix, but the authors motivate it with way too general and vague relationships, such as "determining the number of nodes in neural networks". Somewhat oddly, the problem is highlighted to be of interest in supervised problems even though one would expect it to be much more important in unsupervised ones.

2. Complete lack of details for related work. Methods such as PA and MAP are described with vague one-sentences summaries that tell nothing about how they actually work. There would have been ample space to provide the mathematical formulations.

3. No technical contribution. The proposed method is trivial variant of randomised testing, described with single sentence "Bootstrapped samples R_B are simply generated through random sampling with replacement of the values of R." with literally no attempt of providing any sort of justification why this kind of random sampling would be good for the proposed task or what kind of assumptions it builds on.

4. Poor experiments using really tiny artificial data sets, reported in unprofessional manner (visual style in plots changes from figure to figure, tables report irrelevant numbers in hard-to-read format etc). No real improvement over the somewhat random choice of comparison methods that do not even represent the techniques people would typically use for this problem.

---

### Official Review · AnonReviewer3 · 2017-11-27
**Not ready for publication**

**Rating:** 2
**Confidence:** 5

**Review:**

The authors propose a bootstrap-based test for determining the number of latent dimensions to retain for linear dimensionality reduction (SVD/PCA). The idea is to retain eigenvectors which are larger than a bootstrap average. The resulting approach is evaluated on two simulated datasets (dense and sparse)as compared to common baselines and evaluated. The results suggest improved performance.

The paper addresses an important problem, but does not seem ready for publication:
 - The evaluation only uses simulated data. Ideally, the authors can evaluate the approach on real data -- perhaps using out of sample variance explained as a criterion?
 - There is limited technical novelty. The bootstrap is well known already. The authors do not provide additional insight, or provide any theory justifying the technique.
 - It's not clear if the results are new:
Paper with related discussion: http://jackson.eeb.utoronto.ca/files/2012/10/Jackson1995.pdf
and a blog post:
https://stats.stackexchange.com/questions/33917/how-to-determine-significant-principal-components-using-bootstrapping-or-monte-c

---

### Official Review · AnonReviewer2 · 2017-11-27
**A boostrapped SVD is proposed and contrasted some existing approaches to determining the number of components in PCA. The paper is insufficiently contratested existing literature.**

**Rating:** 3
**Confidence:** 4

**Review:**

The authors propose the use of bootstrapping the data (random sampling entries with replacement) to form surrogate data for which they can evaluate the singular value spectrum of the SVD of the matrix to the singular values of the bootstrapped data, thereby determining the number of latent dimensions in PCA by the point in which the singular values are no greater than the bootstrapped sampled values.  The procedure is contrasted to some existing methods for determining the number of latent components and found to perform similarly to another procedure based on bootstrapping correlation matrices, the PA procedure.

Pros:
Determining the number of components is an important problem that the authors here address.

Cons:
I find the paper poorly written and the methodology not sufficiently rooted in the existing literature. There are many approaches to determining the number of latent components in PCA that needs to be discussed and constrasted including:
Cross-validation:
http://scholar.google.dk/scholar_url?url=http%3A%2F%2Fwww.academia.edu%2Fdownload%2F43416804%2FGeneralizable_Patterns_in_Neuroimaging_H20160306-9605-1xf9c9h.pdf&hl=da&sa=T&oi=gga&ct=gga&cd=0&ei=rjkXWrzKKImMmAH-xo7gBw&scisig=AAGBfm2iRQhmI2EHEO7Cl6UZoRbfAxDRng&nossl=1&ws=1728x1023
Variational Bayesian PCA:
https://www.microsoft.com/en-us/research/publication/variational-principal-components/
Furthermore, the idea of bootstrapping for the SVD has been discussed in prior publications and the present work need to be related to these prior works. This includes:

Milan, Luis, and Joe Whittaker. “Application of the Parametric Bootstrap to Models That Incorporate a Singular Value Decomposition.” Journal of the Royal Statistical Society. Series C (Applied Statistics), vol. 44, no. 1, 1995, pp. 31–49. JSTOR, JSTOR, www.jstor.org/stable/2986193.

Fisher A, Caffo B, Schwartz B, Zipunnikov V. Fast, Exact Bootstrap Principal Component Analysis for p > 1 million. Journal of the American Statistical Association. 2016;111(514):846-860. doi:10.1080/01621459.2015.1062383.

Including the following package in R for performing bootstrapped SVD: https://cran.r-project.org/web/packages/bootSVD/bootSVD.pdf

The novelty of the present approach is therefore unclear given prior works on bootstrapping SVD/PCA.

Furthermore, for sparse data with missing entries there are specialized algorithms handling sparsity either using imputation or marginalization, which would be more principled to estimate the PCA parameters.

Finaly, the performance appears almost identical with the PA procedure. In fact, it seems bootstrapping the correlation matrix has a very similar effect as the proposed bootstrapping procedure. Thus, it seems the proposed procedure which is very similar in spirit to PA does not have much benefit over this procedure.

Minor comments:
Explain what SW abbreviates when introduced first.
We will see that it PA a close relationship with BSVD-> We will see that PA is closely related to BSVD

more effective than SVD under certain conditions (?). – please provide reference instead of ?

But table 4 that shows -> But table 4 shows that

We can sum up with that the result seems ->To summarize, the result seems

---

### Public Comment · (anonymous) · 2017-11-22
**Nice idea!**

I think this is quite a nice idea.

One thing that would make the experiments more complete would be comparison to the bi-crossvalidation method that is sometimes used in stats (statweb.stanford.edu/~owen/reports/AOAS227.pdf). The BCV method can be quite slow for large matrices whereas it seems like the method proposed here can be efficiently parallelized.

---

### Decision · Program_Chairs · 2018-01-29
**ICLR 2018 Conference Acceptance Decision**

**Decision:**

Reject

**Comment:**

The paper addresses the important question of determining the intrinsic dimensionality, but there remain several issue, which make the paper not ready at this point: unclear exposition, lack of contextualisation of existing work and seemingly limited insights. The reviewers have provided many suggestions to improve the paper which we hope will be useful to improve the paper.